# Development of a Highly Efficient Environmentally Friendly Plasticizer

**DOI:** 10.3390/polym14091888

**Published:** 2022-05-05

**Authors:** Irina N. Vikhareva, Guliya K. Aminova, Aliya K. Mazitova

**Affiliations:** Applied and Natural Sciences Department, Ufa State Petroleum Technological University, Mendeleeva St. 195, 450080 Ufa, Russia; aminovagk@inbox.ru (G.K.A.); elenaasf@yandex.ru (A.K.M.)

**Keywords:** adipate, effective plasticizer, environmentally friendly, esterification, polyvinyl chloride, technological

## Abstract

The purpose of this work is the synthesis of adipic acid ester and the study of the possibility of its use as a PVC plasticizer. The resulting butyl phenoxyethyl adipate was characterized by Fourier-transform infrared spectrometry, thermogravimetric analysis (TGA) and differential scanning calorimetry (DSC). The compatibility, effectiveness and plasticizing effect of butyl phenoxyethyl adipate in comparison with dioctylphthalate (DOP) were determined. The new environmentally friendly plasticizer has good compatibility with PVC and high thermal stability. The effectiveness of the plasticizing action of adipate based on the glass-transition temperature was 132.2 °C in relation to pure PVC and 7.7 °C in comparison to compounds based on DOP. An increase in the fluidity of the melt of polyvinyl chloride (PVC) compounds in the temperature range of 160–205 °C by 19–50% confirms a decrease in the energy intensity of the processes of manufacturing and the processing of polymer materials containing a new additive.

## 1. Introduction

Modern polymeric materials are high-quality substitutes for natural raw materials with valuable qualities, many of which are not inherent in natural materials. For this reason, polymeric materials are widely used and are reliably and effectively used in almost every sector of the world economy. The cumulative global production of polymer materials since the 1950s is 8.3 billion tons [1]. Plastic production over the past 13 years has totaled more than 300 million tons annually [2].

It is also important that the use of polymers contributes to the low-carbon development of the economy [3,4]. For example, plastic wrap films are lighter and more packable, which means less energy consumption and a smaller carbon footprint [5]. The amount of carbon dioxide emissions during the production (kg of CO_2_ per 1 kg of products) of an aluminum can is 11.4 and that of PVC is 4.4 [6,7]. Plastic is an element of a circular economy, and its use allows the spending of resources more economically [8,9,10,11,12,13,14].

However, despite the enormous economic effect that is achieved when polymer materials are introduced into industry and everyday life, the environmental safety of their use is of great importance [15,16].

Currently, the composition of polymer materials includes various chemical additives that are systematically released during the manufacture, processing and operation of products, polluting the environment and possibly penetrating into the human body [17,18,19].

To reduce their negative impact, it is necessary to ensure the environmental safety of polymeric materials when creating plastics and when developing technological regulations using non-toxic additives [20,21].

Today, polyvinyl chloride in terms of consumption takes third place after polyethylene and polypropylene; therefore, the development and use of environmentally friendly biodegradable additives for these polymers is relevant, which contributes to solving the urgent problem of environmental pollution with plastic waste [22].

Renewable sources based on plant materials or their production waste to be utilized as fillers for regulating the biodegradability of polymer composite materials are an alternative for the development of economically and environmentally attractive technologies [23]. However, such polymer composite materials are inferior in physical, mechanical, technological and operational characteristics to traditional polymers. For this reason, it is advisable to modify polymer compositions using plasticizers capable of serving as a source of organic substances for microorganism-destructors under ambient conditions [24,25].

The expansion of applications for plasticizers since the 1960s has contributed to the search for new competitive additives [26]. In market conditions, phthalate plasticizers turned out to be the most economical and in demand [27]. According to expert data, in 2019, the capacity of the global market of plasticizers is about 8 million tons [28]. Of these, the phthalate-free market constitutes 35% of the total [29]. Among diester plasticizers, phthalates occupy more than 80% of the market [30].

DOP takes first place in terms of production and consumption [31]. DOP has optimal plasticizing properties at a low cost and is the main standard by which other plasticizers are compared.

However, due to the migration of DOP from polymer products and its identified toxicity, the share of consumption of this plasticizer tends to gradually decrease.

The demand for phthalate-free plasticizers around the world is gradually increasing. Among phthalate-free plasticizers, adipic acid esters are in the lead in terms of consumption [32].

For this purpose, it is possible to use biodegradable plasticizers in the formulations of PVC compositions, for example, esters of adipic acid [33,34]. In numerous works, the biotoxicity and the period of biodegradation of the industrial adipate plasticizer DOA were investigated, and it was shown that this additive for PVC is non-toxic to various types of living microorganisms and that the period of its biodegradation is 6 months [35].

The relevance of the transition to green technologies contributes to the expansion of research on the production of adipic acid from raw materials on a biological basis using chemical and/or biological processes [36,37,38,39]. The industry of end users of adipic acid is increasingly inclined towards the use of bioadipic acid [40].

Plasticizers based on other acids are known and used [41], for example, citrates [42], but the cost of their production is much higher [43].

Esters of succinic acid are a renewable resource, which makes them an attractive promising starting material for industrial production [44]. For this reason, succinates are increasingly replacing terephthalic acid esters [45]. Test results have shown that the plasticizing properties of succinates improve with an increase in the length of the hydrocarbon chain, while the biodegrading properties deteriorate [46,47].

Trimellitates are somewhat inferior to phthalate plasticizers in terms of frost resistance of plastic compounds [48].

The consumption of epoxy plasticizers has increased in recent years due to the availability of vegetable oils [49]. Three oils are mainly used as raw materials for production: soybean, linseed and tall oil [50]. These plasticizers are not fluid, increase resistance to heat and ultraviolet light, have low migration and low toxicity and increase the frost resistance of the material, and, with increases in the content reduce the stability of film properties in a humid atmosphere and sweat with increasing temperature. As a rule, they are used in conjunction with small amounts of low-molecular-weight plasticizer.

Known plasticizer based on castor oil, which is gradually hydrogenated, esterified and acetylated to obtain triacetylated monoglyceride ester. The disadvantage of using castor oil as a feedstock is that products based on it are more expensive than phthalates, and the world supply of castor oil is limited [51].

Another type of natural-product-based plasticizer is acetylated soybean oil [43]. Technology converts epoxy groups in epoxidized oil into esters of vicinally diacetylated fatty acids. The technology can potentially be expanded to produce a variety of similar epoxy plasticizers [52].

Another example of the use of natural raw materials is the creation of bioplasticizers based on sucrose esters, which are also a renewable resource. Their content in molasses, a waste product of sugar beet production, is approximately 45%. In addition to the plasticizing effect, esters of sucrose and fructooligosaccharides have an antimicrobial effect [53]. Sucrose palmitate and glucose hexanoate were studied as bioplasticizers [54]. The esters showed good miscibility with PVC and good plasticization efficiency, as well as good mechanical properties in the form of higher strain at break and a lower modulus of elasticity.

Methods for processing polymer composite materials are characterized by high manufacturability, productivity, a high degree of automation, minimum energy consumption and the ability to manufacture several products of complex shape in one molding cycle [55]. The achievement of the desired characteristics of polymer composite materials is primarily determined by the type, quantity and ratio of components.

Taking into account the above factors, the development of environmentally friendly additives that contribute to the high manufacturability, productivity and minimal energy consumption of the processing and manufacturing of polymer composite materials is relevant.

In addition, it is important that the resulting additives contribute to solving the environmental problem of plastic pollution and render polymers biodegradable.

The introduction of plasticizers into the PVC composition formulation promotes a targeted change in the structure and properties of the polymer, which definitely leads to an improvement in the rheological and physicomechanical characteristics of the resulting compound: strength, frost resistance and brittleness, hardness, melt flow rate, impact strength and manufacturability, as well as thermal, electrophysical and other properties of polymers [56,57]. According to the studies, the manufacturability of PVC composite processing is evaluated according to the rheology of the melts. A fairly reliable and widespread method in practice is to determine the processability of polymers by the value of the melt flow rate (MFR). This indicator allows the establishment of a temperature range for the processing of the polymer composition and provides the necessary conditions for its implementation.

In this regard, when developing new environmentally friendly plasticizers, it is important to determine their effect on the processing of polymer composite materials. In order to create resource- and energy-saving technologies, additives are needed to ensure these characteristics. Therefore, in this work, technologically important characteristics of the developed plasticizers were studied: thermal stability, compatibility and rheology.

## 2. Materials and Methods

### 2.1. Starting Materials

Adipic acid was purchased from Radici Group, Selbitz-Hochfranken, Bavaria, Germany. Butanol and phenol were purchased from The Company «Rearus», Moscow, Russia. Ethylene oxide was purchased from ECOTECH Chemical Components Plant, Moscow, Russia. Sodium hydroxide was purchased from Joint Stock Company “Caustic”, Sterlitamak, Russia, it is a white solid with a main substance content of 98.2%. P-Toluenesulfonic acid was purchased from Component-Reagent, Moscow, Russia, it is a white solid with a main substance content of 95%. Toluene was purchased from Public Joint-Stock Company “Joint-Stock Oil Company Bashneft”, Ufa, Russia. It is a colorless liquid with a characteristic smell and a main substance content of 99%. Suspension polyvinyl chloride (PVC) (Joint Stock Company “Caustic”, Russia, Sterlitamak): industrial samples of PVC 7059M.

### 2.2. Synthesis Methods

#### 2.2.1. Synthesis of Phenoxyethol

A calculated amount of phenol and sodium hydroxide catalyst was loaded into a round-bottomed chemical reactor equipped with a thermometer, magnetic stirrer, reflux condenser and a special device for introducing ethylene oxide into the prepared reaction mass.

The reaction mixture was heated to 130 °C and then purged with nitrogen to remove air. Further, with a working magnetic stirrer, the prepared ethylene oxide was gradually introduced. The required ethylene oxide feed rate was adjusted accordingly, so that the unreacted oxide condensed in the reflux condenser and was returned to the chemical reactor without flooding. After feeding all of the required amount of ethylene oxide, the temperature of the reaction mixture was maintained in the specified range for another 1–1.5 h and then gradually cooled to room temperature.

The catalyst was neutralized with a calculated amount of sulfuric acid, and the resulting mass was filtered. Then a fraction was distilled from the reaction mixture, boiling at 50 °C at 10 mm Hg.

#### 2.2.2. The Synthesis of Butyl Phenoxyethyl Adipate

Butyl phenoxyethyl adipate was obtained by the sequential esterification of adipic acid. Initially, solvent (150 mL), adipic acid (146 g, 1 mol) and alcohol (1 mol) were loaded into the reactor in a 1:1 ratio. The the heating was turned on. The required amount of catalyst was added (1% by weight). The reaction mass was bubbled with an inert gas. The heating of the reaction mixture continued for 1 h. Phenoxyethanol (1.2 mol) was then added, and heating continued for another 2 h. The reaction was monitored by the acid number of the esterificate and the amount of released water. The reaction mixture was cooled, and the target ester was isolated.

### 2.3. Methods of Analysis

#### 2.3.1. Analysis of Physicochemical Parameters of Butyl Phenoxyethyl Adipate

An analysis of physicochemical characteristics was carried out in accordance with the regulatory requirements for plasticizers [58,59]. For this, the following indicators were determined: acid number, ester number and density.

#### 2.3.2. Characterization of Butyl Phenoxyethyl Adipate

An analysis was performed by FTIR spectroscopy on an FTIR-8400S FTIR spectrometer (Shimadzu, Kyoto, Japan). For this, KBr tablets were prepared according to the standard procedure. The IR absorption spectra of the target product were recorded in the range 3700–450 cm^−1^ at room temperature. The resolution was 4 cm^−1^, and the number of scans was 20.

#### 2.3.3. High-Performance Liquid Chromatography

The study of the resulting product was carried out using HPLC (LC-10 from SHIMADZU, Kyoto, Japan) in reverse phase mode. A model refractometric detector (RIDK 101, Prague, Czech Republic) was used as a detector. To separate the components, columns (150 × 4.6 mm) filled with Separon-C18 sorbent (particle size 5 μm) were taken. Separation was carried out in an acetonitrile-water eluent system taken in a 67/33 volumetric ratio. The flow rate of the eluent was 0.5 mL/min. The volume of injected samples was 10 μL. Quantitative analysis was performed using the absolute calibration method. Calibration solutions contained adipic acid, alcohols and esters.

#### 2.3.4. Determination of PVC Compatibility with Plasticizer

This indicator was evaluated by the critical temperature of thedissolution of the synthesized ether in polyvinyl chloride. For this, 0.5 g of polymer was mixed with 5 g of plasticizer, and the mixture was gradually heated at a rate of 2 °C per minute, and the change in the appearance of the contents of the test tube was visually determined. The critical dissolution temperature was taken as the temperature at which PVC was completely dissolved in the studied plasticizer and the solution became transparent. The indicator was calculated as the average value of at least three measurements.

#### 2.3.5. Thermogravimetric Analysis of Butyl Phenoxyethyl Adipate

Themoanalytical studies were carried out on a TGA/DSC-1 thermal analysis device (Mettler Toledo, Uster, Switzerland).

The thermal stability of the product was investigated in a dynamic mode at a constant heating rate of 5 deg/min in the temperature range from 25 to 500 °C in air. For the experiment, the weight of the weighed portion of the sample was 5–10 mg. Crucibles with a volume of 100 μL were made of aluminum oxide. The results were processed using a computer.

#### 2.3.6. Differential Scanning Calorimetry of Butyl Phenoxyethyl Adipate

DSC analysis of the product was performed on a DSC-1 instrument (Mettler Toledo, Uster, Switzerland).

The analysis was carried out in the temperature range from −50 to 150 °C in air in a dynamic mode with a constant heating/cooling rate of the sample—10 deg/min. To carry out the analysis, crucibles made of aluminum with a volume of 40 μL were used, which with a sample weighing 4–8 mg was sealed with a lid using a press. The results were processed using a computer. The final cooling temperature of the sample was −100 °C.

#### 2.3.7. Determination of Glass Transition Temperature

The glass transition temperatures of PVC composites containing the developed plasticizers were determined by differential scanning calorimetry on a device DSC-1 (Mettler Toledo, Uster, Switzerland).

The glass-transition temperature was determined by DSC on a DSC-1 instrument (Mettler Toledo) in dynamic mode at a constant heating rate of 2 K/min. The analysis was carried out in the temperature range from –100 to 100 °C in air. The mass of the sample taken for measurements was 4–8 mg. For the analysis, aluminum crucibles with a volume of 40 μL were used. A weighed portion of the sample was placed in a crucible and sealed with a lid using a press. After quenching at 90 °C for 5 min, the sample was heated to 100 °C at a rate of 2 °C/min. The first heating cycle was used to remove any heat history. The glass-transition temperature of the polymer was determined from the DSC curve obtained in the second heating cycle of the sample using the supplied software. Using the tangent method, the middle of the bend (step) on the curve was determined, which was taken as the glass-transition temperature.

To assess the effect of the synthesized ether on the glass-transition temperature, PVC composition were made: 50 parts by weight of plasticizer per 100 parts by weight of PVC. The composition of the plasticizer was, in ppm, DOP: 83.3 and adipate plasticizer: 16.7.

#### 2.3.8. Determination of the Melt Flow Rate

The melt flow rate (MFR) was estimated by capillary viscometry using an IIRT-AM plastometer (International Standard 1133-1:2011(E)). The MFR value corresponds to the mass of the polymer in grams flowing out of the capillary (capillary 8 mm long, 2.09 mm in diameter) of the device in 10 min at a certain temperature and pressure. The MFR of PVC composites was measured in the temperature range 160–205 °C at a load of 49N. A total of 4–5 g of the investigated PVC composition was introduced into the device heated to a predetermined temperature and kept under pressure for 5 min, then the capillary was opened, carrying out the outflow of the composition melt.

To measure the MFR parameter, at least five extruded segments of the composite were used, cut off at certain equal time intervals. The mass of the obtained extruded sections with an error of not more than 0.0001 g was measured after they were cooled. The value of the MFR parameter was calculated by the Equation (1):(1)MFR=(m*600)/t
where *m*—average mass of extruded segments, g; *t*—time interval between two consecutive cutoffs of segments, sec.

## 3. Results

### 3.1. Synthesis of Ethoxylated Alcohols

Phenoxyethanol was obtained by the reaction of ethylene oxide with phenol at an equimolar ratio.

In appearance, ethoxylated phenol of the composition C_6_H_5_O(CH_2_CH_2_O)H is a transparent liquid. Subsequently, phenoxyethanol was used to synthesize the target ester. Phenoxyethanol is a colorless oily liquid. The yield was 89% of the theoretical maximum.

The phenoxyethanol had a degree of ethoxylation *n* = 1: density—1.1007, refractive index—1.5314, molecular weight (calculated)—138, reaction time—1.3 h.

### 3.2. The Synthesis of Phenoxyethyladipate

The synthesis was carried out according to Figure 1.

The resulting ether is the light clear liquid.

The yield of butyl phenoxyethyl adipate was 279.2 g (86.7% of the theoretical maximum).

The physicochemical properties of butyl phenoxyethyl adipate are presented in Table 1.

### 3.3. IR Spectra

In the synthesized compound, the stretching vibrations of the carbonyl group are shifted to the high-frequency region and are reflected in the spectrum as a strong characteristic band in the region of 1735 cm^−1^ (Figure 1). There are absorption bands characteristic of the vibrations of the C–O–C ester fragment in the region of 1156 cm^−1^.

The spectrum of butyl phenyloxyethyl adipate lacks an absorption band in the range of 1685–1687 cm^−1^, which is characteristic of the stretching vibrations of the carbonyl group in associates of aliphatic carboxylic acids.

### 3.4. Determination of the Compatibility of Butyl Phenoxyethyl Adipate with PVC

For the plasticizer butyl phenoxyethyl adipate, which is a free-flowing solid product, the studies were carried out using a mixture as a plasticizer in the following ratio: DOP at 5 parts by weight and adipate ester at 1 part by weight. When determining the critical dissolution temperature, 0.05 g of PVC was dissolved in 5 mL of the mixture. The research results are shown in Table 2.

The compatibility of the plasticizer with PVC resin was determined by the formula:(2)A=TDOP/Tcr·100%,
where *TDOP* is the critical temperature of dissolution of the plasticizer in DOP, and *Tcr* is the critical temperature of the dissolution of PVC resin in a mixture of the investigated plasticizer with DOP.

### 3.5. TGA of Butyl Phenoxyethyl Adipate

In the course of thermal analysis, a sample of an industrial plasticizer, dioctyl phthalate, was used as a comparison.

To assess the thermal stability of the synthesized adipate according to the TGA thermogram, the following parameters were determined (Figure 2, Table 3):

*T_b_*—temperature corresponding to the beginning of the decrease in the mass of the sample upon heating;

Δ*m*_180_—reduction of the sample weight when heated to a temperature of 180 °C, corresponding to the temperature range of processing PVC compositions;

*T_max_*—temperature of the maximum decomposition rate of the sample;

*T_d_*—decomposition temperature.

### 3.6. DSC Analysis of Butyl Phenoxyethyl Adipate

In the course of thermal analysis, a sample of an industrial plasticizer dioctyl phthalate was used as a comparison.

Melting and crystallization temperatures were determined for butyl phenoxyethyl adipate, which is a solid product, from the DSC curves shown in the thermograms (Figure 3, Table 4).

### 3.7. Determination of the Glass-Transition Temperatures of PVC Compositions

The transition of a polymer from a glassy to a highly elastic state is accompanied by an increase in the heat capacity of the polymer, which is reflected in the form of a characteristic break (step) on the DSC curve (Figure 4). The glass-transition temperature was found by the tangent method along the middle of the step corresponding to this transition.

Table 5 shows the results of a study of the effect of plasticizers on the glass-transition temperature of PVC compositions.

### 3.8. Determination of the Rheological Characteristics of PVC Composition with BPEA

To assess the rheological properties of plasticized PVC composites, we used the melt flow rate (MFR) of the polymer, which is widely used in practice to characterize the processability of polymer materials and select a processing method. The studies were carried out on an IIRT-AM device in the temperature range of 160 to 205 °C.

The rheological properties of PVC-compositions containing mixtures of the DOP plasticizer with the developed adipate of various compositions were studied. When preparing the PVC composition, a mixture of plasticizer was used in the ratio of DOP:adipate 5:1 (wt.).

Initially, the effect of temperature on the MFR of PVC compositions of the following composition (ppm) was investigated: PVC: 100, plasticizer: 100 and tribasic lead sulphate: 1. The composition of the plasticizer was, in ppm: DOP of 83.3 and adipate plasticizer of 16.7. The used load was 49N (Figure 5).

Then the rheological characteristics of the PVC composition were studied (ppm): PVC: 100, plasticizer: 50 and tribasic lead sulphate: 1. The composition of the plasticizer, in ppm, was DOP of 83.3 and adipate plasticizer of 16.7. The used load was 49 N (Figure 6).

We also studied the rheological characteristics of the PVC composition of the composition, in ppm: PVC of 100, plasticizer of 39 and tribasic lead sulphate of 1. The composition of the plasticizer was, in ppm: DOP of 83.3 and adipate plasticizer of 16.7. The used load was 49N (Figure 7).

It has been experimentally established that, with an increase in temperature and plasticizer content, the melt fluidity of PVC composites, regardless of their composition, increases, and this dependence is close to exponential (Figure 5, Figure 6 and Figure 7).

## 4. Discussions

### 4.1. Synthesis of Ethoxylated Alcohols

The literature describes the use of phenol derivatives for the plasticization of polyamides and cellulose acetate, but there are very few data on the possibility of using them as raw materials in the production of polyvinyl chloride plasticizers [60].

The synthesis of ethoxylated alcohols, including phenoxyethanol, is well studied and carried out on an industrial scale [61]. The ethoxylation of phenol was carried out according to well-known methods by the interaction of alcohol with ethylene oxide at a temperature of 120 °C in the presence of a sodium hydroxide catalyst.

The reaction proceeds according to the S_N_^2^ mechanism and is a series-parallel addition of ethylene oxide to phenol.

The composition of the products of the ethoxylation reaction depends on the molar ratio of alcohol and ethylene oxide in the reaction mass. With an increase in the content of ethylene oxide, the degree of ethoxylation of the resulting alcohols increases.

Then the resulting phenoxyethanol was used for the synthesis of butyl phenoxyethyl adipate.

### 4.2. The Synthesis of Butyl Phenoxyethyl Adipate

Initially, the reaction flask was charged with adipic acid, solvent and alcohol butanol. The reflux of the reaction mixture was maintained until the required amount of water was released in the Dean–Stark trap. Then the alcohol phenoxyethanol was added with a slight excess relative to the acid.

### 4.3. High-Performance Liquid Chromatography

Undoubtedly, at the time of the addition of the second alcohol, the reaction mass contained monoester, dibutyl adipate and a small amount of unreacted adipic acid. When phenoxyethanol was added, transesterification of the previously formed diester was unambiguously observed, which is known from the literature on organic synthesis. Thus, the symmetric adipate was converted into the target unsymmetrical ester. The chromatograms obtained by HPLC have two peaks: one peak corresponds to symmetric, and the other to asymmetric.

### 4.4. Analysis of Physicochemical Parameters of Butyl Phenoxyethyl Adipate

Butyl phenoxyethyl adipate was not separated from the symmetrical one. The obtained product was used for testing, and physicochemical analyses showed the compliance of the product with regulatory requirements for use as a plasticizer for polyvinyl chloride.

### 4.5. Determination of the Compatibility of Butyl Phenoxyethyl Adipate with PVC

The compatibility of PVC resin with a plasticizer is one of the most important factors determining the possibility of its use in the development of a plasticized material formulation.

The plasticizer can be gradually released from the composite material by extraction, migration or evaporation. Over time, even a very low vapor pressure of the introduced plasticizer appears [62]. Therefore, the migration rate of the plasticizer from the polymer composition doubles with an increase in temperature by 7 °C [63]. The vapor pressure of the plasticizer used is for guidance only.

The migration of a plasticizer from a place with a higher concentration to a place with a lower concentration is determined by the nature of the polymer [63]. In this case, the less the interaction of the plasticizer with PVC, the lower the limit of compatibility of the plasticizer with PVC and the higher the migration value [64]. At plasticizer concentrations in PVC compositions up to 10%, the plasticizer is completely solvated by the polymer, which leads to a decrease in the detachment of the plasticizer molecules from the polymer [65].

With an increase in the amount of introduced plasticizer from 0 to 10%, migration increases insignificantly [66]. A further increase in the content of the plasticizer in the PVC composite from 15 to 35% leads to an almost linear dependence of the amount of migration on the amount of plasticizer [67]. Thus, with an increase in the content of the plasticizer, the number of fragile bonds in the polymer–plasticizer system increases, which contributes to an increase in migration [68]. The determination of the compatibility of the plasticizer with the polymer makes it possible to purposefully regulate its structure and the operational properties of the resulting material, in order to importantly prevent high volatility and the likelihood of migration to the surface of products, which negatively affects operational characteristics [69].

The determination of the compatibility of the new ethoxylated alcohol adipate with PVC was carried out according to the critical dissolution temperature of PVC resin and compatibility parameter A (%) [70]. For well-compatible plasticizers, this parameter should be greater than 100%.

The results obtained showed that the developed plasticizer butyl phenoxyethyl adipate is characterized by a high ability to dissolve polyvinyl chloride.

### 4.6. Thermal Analysis of Butyl Phenoxyethyl Adipate

To study the possibility of the practical application of the plasticizer in the composition of PVC composites, the following indicators are important: thermal stability, melting and crystallization temperatures, enthalpy of melting and crystallization.

Thermoanalytical studies have shown that the developed adipate has a higher thermal stability in comparison with DOP. The introduction of the developed additive into the PVC composition provides a higher thermal stability during processing, which improves the manufacturability of the compounds.

### 4.7. Determination of Glass-Transition Temperatures of PVC Compositions

At the next stage, the efficiency of the plasticizing action of the new plasticizer was assessed.

The chemical structure of polymer chains and their mutual arrangement determine the physical properties of polymers. These properties of polymers, in particular the glass-transition temperature, are changed by introducing special substances—plasticizers—into the polymer composition during processing. The shift of the glass-transition temperature to the region of lower temperatures is commonly called plasticization. The introduction of a plasticizer into a polymer composition is important from a practical and theoretical point of view. Lowering the glass-transition temperature of the polymer with the introduction of a plasticizer makes it possible to expand the temperature range of its highly elastic state, that is, to increase its frost resistance. In addition, as a result, a decrease in the viscosity of polymer melts makes it possible to significantly facilitate their processing. It is especially important to lower the glass-transition temperature and pour point during the processing of polymers, in which these characteristics are close to or even higher than their decomposition temperature.

A decrease in the glass-transition temperature with the introduction of a plasticizer is an important criterion for assessing the effectiveness of its plasticizing action.

The results shown in Table 2 show that the sample of unplasticized polyvinyl chloride corresponds to Tgt = 87.5 °C. The introduction of polyvinyl chloride 50 ppm plasticizer DOP led to a decrease in the glass-transition temperature of the polymer to −37 °C, that is, by 124.5 °C. A significant decrease in the Tgt parameter was observed when 50 ppm is introduced into the polymer. The synthesized adipate totaled 133.2 °C.

### 4.8. Determination of the Rheological Characteristics of PVC Composition with BPEA

Previously, the chemical nature of the plasticizer was not given due attention in the development of composites. It has now been proven that the plasticizing effect of the developed additive depends on the chemical structure (the nature of the atoms included in the molecule, the polarity of the bonds and the size and shape of the plasticizer molecules) and the ability of the molecule to undergo conformational changes, as well as on the thermodynamic affinity of the plasticizer for the polymer.

A comparison of the rheological characteristics of PVC compositions based on the developed BPEA plasticizer with similar compounds containing DOP showed an increase in melt flow in the temperature range (Figure 5, Figure 6 and Figure 7), which indicates the high efficiency of the plasticizing action of the developed plasticizer, which is superior to the commercial DOP plasticizer. In this regard, this ester contributes to resource and energy efficiency when used as a PVC plasticizer.

## 5. Conclusions

The article describes the successful production of a new promising plasticizer for polyvinyl chloride, butyl phenoxyethyl adipate, by a method of sequential azeotropic esterification in one reaction vessel. The structure of the obtained compound was confirmed by FT-IR spectroscopy. Studies of the compatibility of the obtained compound with PVC showed the possibility of using it as a plasticizer: the critical temperature of polymer dissolution in ether is 101 °C compared to that for DOP, which is 112.5 °C. Thermal analysis by the TGA and DSC methods confirmed the higher thermal-oxidative stability of BPEA in comparison with DOP; the temperature of the maximum decomposition rate is 321 °C and 285 °C, respectively. The test results showed that the use of butyl phenoxyethyl adipate in the composition of the PVC composition provides a significant decrease in the glass-transition temperature of the polymer, by 132.2 °C compared to pure PVC and by 7.7 °C compared to DOP-based compounds. The study of the rheological characteristics of PVC compounds containing butyl phenoxyethyl adipate showed that the developed plasticizer has a pronounced plasticizing effect in relation to PVC, which consists of a significant increase in the melt flow of PVC compounds in the temperature range of 160–205 °C. So, for example, when the content of BPEA in the amount of 16.7 ppm per 100 ppm including PVC, the melt flow rate of the composition at 175 ° C increases by 50%; when the content of BPEA in the amount of 8.35 ppm at 200 °C, it increase by 20%; when the content of BPEA in the amount of 6.5 ppm at 205 °C, it increases by 19%.

Thus, studies have shown that the new plasticizer is characterized by high technological indicators—the thermal stability and efficiency of the plasticizing action—and also contributes to an increase in the fluidity of the melt of polymer compositions and the expansion of the temperature range for processing compounds based on it. This paper also shows that the use of a new environmentally friendly plasticizer contributes to the resource and energy efficiency of PVC compound processing technologies with its content.

## Data Availability

The data presented in this study are available on request from the corresponding author.

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
