# Peer review of "Development of a Highly Efficient Environmentally Friendly Plasticizer"

_polymers, 2022, doi:10.3390/polym14091888_

Round 1

Reviewer 1 Report

  1. The authors cover the background in the introduction. However, it lacks of why and how the experiment being designed.
  2. In figure 1, the authors provide the IR spectrum of the butyl phenoxyethyl adipate. I suggest to mark the corresponding absorption bands in the figure that would help readers to identify the characteristics. 
  3. The figures the authors provides are in different styles. For example figure 4 and figure 5. They have different world styles and sizes. This issue should be corrected. 
  4. Also, in figure 5, 6, and 7. What are the I and II? The condition of each should be marked in the figures.
  5. The authors demonstrate the Tg temp values with PVC, PVC+DOP, and PVC+DOP+BPEA. How about the one with only PVC + BPEA? and the BPEA itself? 
  6. In Figure 3, the authors measure the DSC of BPEA. Does the author measure the DOP since the authors mentions DOP many time in the manuscript? What's the difference in DSC?

Reviewer 2 Report

After review the manuscript there are several questions that need to be improved before the manuscript can be accepted, they are described following:

-please homogenize  the use of abbreviations, due in some parts of manuscript use them but in other write the whole word, for instance: PVC, Dip, BPEA, etc, first time an abbreviation is written must be defined: DSC, TGA, PVC, etc

-In line 39 write "Currently, there is a wide range of polymeric materials, but the sentence has not sense.

-in Section 2.3.2 the correct way to report the FTIR analysis must be 3700-450 cm-1, and please delete the slash before 4cm-1 in resolution and 20 scans number to avoid confusion.

-In section 2.3.6 please indicate the temperature for final cooling process in DSC.

-I recommend that for melt flow rate indicate the ASTM or ISO method followed.

-Sections 3.1 and 3.2 do not be placed in results section, these correspond to experimental section.

-For indicate the yield of BPEA I recommend to use the percentage instead of mass.

-The physicochemical properties of BPEA reported in table 1, are those results from technical sheet or determinate? if the last one is the correct, please indicate how those results were obtained.

-For FTIR spectrum reported in figure 1, I recommend to delete the wavenumber for all the peaks and only insert the characteristic peaks that are described in text. Also I recommend to change to transmittance mode instead of absorbance, this because usually absorbance mode is used when concentration is determinate. Please, delete the text in bottom of figure (green text), and I recommend to correct the baseline in spectrum.

-For all the temperatures reported in DSC and TGA, I recommend to delete the decimals after point due these are not representative for results.

-for Tcr reported in table 2, please indicate how this value was obtained. in equation 1 what is it mean A? please define it.

-For figure 2, please enlarge them and indicate in Y axis the variable that is plotted, I mean is mass, mass%, heat flow, derivative etc, and indicate in caption for easy identification of each of them. the same observation for figure 3, furthermore that text font in those figures are too small, please enlarge them. For figure 4 (DSC thermogram) please indicate heat flow instead of mW in Y axis.

-The text in lines 327-328 must be placed in experimental section, due it is a description of how the test was carried out. In the same paragraph, indicate that various compositions were studied, however only one combination of DOP and BPEA was reported.

-For figures 5 to 7, please indicate what is it mean I, II symbols for plasticizers.

-In section 4.3 report HPLC, but there is not explanation of how that analysis was donde and I recommend to include HPLC chromatogram that is discusses in that section.

-Why report in section 6 a Patent? if this was used a bibliography must be inserted in references section.

Round 2

Reviewer 1 Report

The revised version looks fine to me.

Author Response

English edited

Reviewer 2 Report

After review the corrected version, this shows an improvement, but still there are some corrections that need to be done, specially for figures presented in results section, I still recommend that fig 1 must be corrected as previously was detailed, also because there is not possible to get an Absorbance with negative value, and the wavenumbers of each peak in spectrum must be deleted due the image looks too heaped, all those changes can be donde in labs solutions IR Software or may be if the data are exported to excel or origin the figure can be plotted. The same observation for TGA, and DSC thermograms (figures 2, 3 and 4), which need to be corrected as previously was indicated.Usually the manipulation of data can be done in Technique Software of manufacturer.

Round 3

Reviewer 2 Report

The authors. made the suggested corrections so I recommend to accept the manuscript for publication